# Utility of Neutrophil-to-Lymphocyte Ratio, Platelet-to-Lymphocyte Ratio, and Systemic Immune Inflammation Index as Prognostic, Predictive Biomarkers in Patients with Metastatic Renal Cell Carcinoma Treated with Nivolumab and Ipilimumab

**DOI:** 10.3390/jcm10225325

**Published:** 2021-11-16

**Authors:** Koji Iinuma, Torai Enomoto, Kei Kawada, Shota Fujimoto, Takashi Ishida, Kimiaki Takagi, Shingo Nagai, Hiroki Ito, Makoto Kawase, Chie Nakai, Kota Kawase, Daiki Kato, Manabu Takai, Keita Nakane, Koji Kameyama, Takuya Koie

**Affiliations:** 1Department of Urology, Gifu University Graduate School of Medicine, Gifu 5011194, Japan; kiinuma@gifu-u.ac.jp (K.I.); buki2121@gifu-u.ac.jp (M.K.); chie_na@gifu-u.ac.jp (C.N.); stnf55@gifu-u.ac.jp (K.K.); andreas7@gifu-u.ac.jp (D.K.); takai_mb@gifu-u.ac.jp (M.T.); keitaco@gifu-u.ac.jp (K.N.); 2Department of Urology, Matsunami General Hospital, Hashima-gun 5016062, Japan; try@mghg.jp; 3Department of Urology, Gifu Prefectural General Medical Center, Gifu 5008717, Japan; keinedvedon@yahoo.co.jp; 4Department of Urology, Ogaki Municipal Hospital, Ogaki 5038502, Japan; uro2@omh.ogaki.gifu.jp; 5Department of Urology, Gifu Municipal Hospital, Gifu 5008513, Japan; justaskaxis@gmail.com; 6Department of Urology, Daiyukai Daiichi Hospital, Ichinomiya 4918551, Japan; kimiaki_takagi5619@yahoo.co.jp; 7Department of Urology, Toyota Memorial Hospital, Toyota 4718513, Japan; shingo-nagai@nifty.com (S.N.); seanoel2@gmail.com (H.I.); 8Department of Urology, Kizawa Memorial Hospital, Minokamo 5058503, Japan; i2001029@yahoo.co.jp

**Keywords:** nivolumab, ipilimumab, renal cell carcinoma, neutrophil-to-lymphocyte ratio, plate-let-to-lymphocyte ratio, systemic immune inflammation index

## Abstract

The aim of this study was to assess the utility of neutrophil-to-lymphocyte ratio (NLR), plate-let-to-lymphocyte ratio (PLR), and systemic immune inflammation index (SII) as predictive biomarkers with oncological outcomes for metastatic renal cell carcinoma (mRCC) patients treated with nivolumab and ipilimumab (NIVO + IPI). We conducted a retrospective multicenter cohort study assessing patients with mRCC treated with NIVO + IPI at eight institutions in Japan. In this study, the follow-up period was median 14 months. The 1-year overall- and progression-free survival (PFS) rates were 89.1% and 63.1, respectively. The objective response rate (ORR) and disease control rate (DCR) were 41.9% and 81.4%, respectively. The 1-year PFS rates were 85.7% and 49.1% for NLR ≤ 2.8 and >2.8, respectively (*p* = 0.005), and 75.5% and 49.7% for PLR ≤ 215.6 and >215.6, respectively (*p* = 0.034). Regarding SII, the 1-year PFS rates were 90.0% and 54.8% when SII was ≤561.7 and >561.7, respectively (*p* = 0.023). Therefore, NLR, PLR, and SII levels in mRCC patients treated with NIVO + IPI may be useful in predicting oncological outcomes.

## 1. Introduction

To date, the recommended agents as a systemic first-line treatment for metastatic renal cell carcinoma (mRCC) have been tyrosine kinase inhibitors (TKIs) and vascular endothelial growth factor (VEGF) inhibitors, including sunitinib and pazopanib [1]. However, the treatment strategy was dramatically altered following the advent of immune checkpoint inhibitors (ICIs), such as nivolumab (NIVO), a programmed cell death protein 1 (PD-1) inhibitor, ipilimumab (IPI), and anti-cytotoxic T-lymphocyte antigen 4 (CTLA-4) monoclonal antibody [2]. A phase III trial compared patients who were previously treated for advanced RCC (CheckMate 025 [3]) with NIVO and everolimus; patients treated with NIVO had a significantly longer overall survival (OS) compared with those treated with everolimus (*p* = 0.0018). In other recent randomized phase III trials (Checkmate 214 [4], KEYNOTE-426 [5], JAVELIN Renal 101 [6], IMmotion151 [7], and CheckMate 9ER [8]), combining ICI and/or TKI therapies reportedly affords greater clinical benefits, especially in terms of oncological outcomes, than NIVO or TKI monotherapy in mRCC [9,10]. In the final oncological outcomes of Checkmate 214 after the long-term follow-up period, NIVO + IPI continued to demonstrate durable efficacy benefits compared with sunitinib [11]. Massari et al. reported that the clinical benefit obtained by patients with mRCC who received immunotherapy combinations with a complete response (CR) rate was more than in those who received the TKI and ICI combination [12].

However, treatment with immune-oncology (IO) agents might not confer equivalent clinical benefits in all patients with mRCC. Several studies have reported that the tumor tissue expression of anti-programmed death ligand-1 (PD-L1) may be associated with oncological outcomes after IO therapy for various malignant neoplasms [13,14]; however, the utility of PD-L1 expression as a prognostic factor for mRCC remains controversial [3,4,5]. In addition, several studies have suggested a correlation between baseline plasma levels of soluble PD-1, PD-L1, BTN3A1, and PBRM1 in response to NIVO in patients with mRCC [15,16]. However, systemic inflammation contributes to the progression of malignant neoplasms [17]. It can be relatively easy to measure the neutrophil-to-lymphocyte ratio (NLR) and platelet-to-lymphocyte ratio (PLR) using peripheral blood counts. Several studies have explored the role of these parameters, focusing on their potential as biomarkers for predicting outcomes of patients treated with NIVO for various types of solid cancer [17,18,19]. Although higher NLR and PLR have been associated with treatment failure and increased risk of death, lower NLR following NIVO treatment reportedly improves oncological outcomes [18,19]. Similarly, the systemic immune inflammation index (SII), calculated as neutrophils × platelets/lymphocytes, has shown a significant association with prognosis in several malignant tumors, including mRCC [20,21,22,23].

To date, the utility of NLR, PLR, and SII as prognostic biomarkers for predicting oncological outcomes in patients with mRCC receiving NIVO + IPI remains unclear. Accordingly, we aimed to evaluate whether NLR, PLR, and SII can predict oncological outcomes, especially progression-free survival (PFS), in patients with mRCC who received NIVO + IPI.

## 2. Materials and Methods

### 2.1. Patients

This study was approved by the Institutional Review Board of Gifu University (approval number: 2020-271) and respective institutional review boards. Patient consent was not required owing to the retrospective nature of the study. The provisions of the ethics committee and the ethics guidelines in Japan did not require written consent since the study information is disclosed to the public in case of retrospective and/or observational studies using material such as existing documentation. Details of the study can be accessed at http://www.med.gifu-u.ac.jp/file/2020-271.pdf, accessed on 14 October 2021.

We conducted a retrospective multicenter cohort study in patients with mRCC treated with NIVO + IPI at eight institutions in Japan between August 2018 and March 2021. All patients were stratified into intermediate- or poor-risk groups according to the International Metastatic Renal Cell Carcinoma Database Consortium (IMDC) risk model [24]. Patients previously treated with TKIs, VEGF, or mammalian target of rapamycin pathway inhibitors and those with missing relevant data were excluded. The collected clinicopathological data included age, sex, body mass index, Eastern Cooperative Oncology Group performance status [25], IMDC risk group, histology, neutrophil count, lymphocyte count, thrombocyte count, NLR, PLR, SII, surgical history, number of metastases, and metastatic site. Blood cell counts were performed at baseline (within a week before NIVO + IPI).

### 2.2. Treatment Schedule

Before September 2018, 3 mg/kg NIVO and 1 mg/kg IPI were intravenously administered to mRCC patients at 3-week intervals. Induction therapy comprised four cycles of NIVO + IPI, followed by maintenance therapy with 3 mg/kg NIVO at 2-week intervals. After October 2018, induction therapy with 240 mg NIVO was applied, followed by maintenance therapy with 3 mg/kg or 240 mg/body weight NIVO at 2-week intervals. The treatment regimen was continued until radiologically proven disease progression or intolerance to treatment-related adverse events (TRAEs).

### 2.3. Patient Evaluation

The following baseline parameters were evaluated: the complete medical history, physical findings, and chest, abdominal, and pelvic findings on computed tomography (CT) and magnetic resonance imaging (MRI). Moreover, the American Joint Committee on Cancer Staging Manual [26] was used to determine tumor stage.

In all patients, CT or MRI was performed at 1- to 3-month intervals until disease progression was radiologically proven, or treatment was discontinued due to TRAEs. Based on the Response Evaluation Criteria in Solid Tumors (RECIST) guidelines (version 1.1) [27], the best overall response was determined as complete response (CR), partial response (PR), stable disease (SD), or progressive disease (PD). Objective response rate (ORR) was defined as the proportion of patients that achieved CR or PR, and disease control rate (DCR) was defined as the proportion of patients with CR, PR, or SD.

NLR and PLR were calculated by dividing the absolute neutrophil and absolute platelet counts, respectively, by the absolute lymphocyte count within the peripheral blood. SII was calculated using the following formula: the absolute platelet count × the absolute neutrophil count/the absolute lymphocyte count. Based on the area under the receiver operating characteristic (ROC) curve [28], the cutoff values for NLR, PLR, and SII were defined as the minimum value for (1 − sensitivity)^2^ + (1 − specificity)^2^. Patients were grouped according to age, as defined by median values. In addition, the patients were divided into two groups with TRAEs ≥ Grade 3 and ≤Grade 2.

### 2.4. Safety

Based on the National Cancer Institute Common Terminology Criteria for Adverse Events (version 5.0) [29], we evaluated TRAEs until at least 100 days after the last administration of NIVO + IPI, beginning from the initiation of treatment.

### 2.5. Statistical Analysis

The primary endpoint of this study was PFS. The secondary endpoints were ORR, DCR, and OS. JMP 14 software (SAS Institute Inc., Cary, NC, USA) was used for data analysis. The follow-up duration was defined as the interval from the date of NIVO + IPI initiation to the last follow-up examination or the documented date of death, whichever occurred first. OS was defined as the interval between treatment initiation and death. PFS was defined as the interval from treatment initiation to the first RECIST-defined disease progression or death, whichever occurred earlier. OS and PFS were estimated using the Kaplan-Meier method. A two-sided 5% significance level was used for all statistical inferences.

## 3. Results

### 3.1. Patients

The pretreatment characteristics of the 43 patients from eight institutions in Japan are listed in Table 1. In this study, the follow-up period was median 14 months (interquartile range (IQR), 7.5–23.5 months). The median number of NIVO + IPI cycles during the induction phase was 4 (IQR, 3–4), while that of NIVO during the maintenance phase was 0 (IQR, 0–13.8). The median values were 3.5 for NLR, 215.6 for PLR, and 1045.2 for SII. The ROC analysis showed that the cutoff values were 2.8 (sensitivity, 89.5%; specificity, 45.8%) for NLR, 215.6 (sensitivity, 68.4%; specificity, 62.5%) for PLR, and 561.7 (sensitivity, 100%; specificity, 37.5%) for SII.

### 3.2. Efficacy and Oncological Outcomes

Four patients (9.3%) achieved CR, 14 (32.6%) achieved PR, and 17 (39.5%) showed SD. The ORR and DCR were 41.9% and 81.4%, respectively.

The OS rate was 94.7% at 6 months and 89.1% at 12 months (Figure 1a). The PFS rate was 76.1% at 6 months 63.1% at 12 months (Figure 1b). This study did not determine the median OS and PFS. No correlation existed between PFS and age, sex, or IMDC risk classification (*p* = 0.651, *p* = 0.760, or *p* = 0.534). Additionally, TRAEs were not significantly associated with PFS (*p* = 0.085). The 1-year PFS rates were 85.7% and 49.1% for NLR ≤ 2.8 and >2.8, respectively (*p* = 0.005; Figure 2a), and 75.5% and 49.7% for PLR ≤ 215.6 and >215.6, respectively (*p* = 0.034; Figure 2b). Regarding SII, the 1-year PFS rates were 90.0% and 54.8% when SII was ≤561.7 and >561.7, respectively (*p* = 0.023; Figure 2c).

### 3.3. Safety

TRAEs are listed in Table 2. Twenty patients (46.5%) were administered high-dose glucocorticoids (≥40 mg prednisone per day or equivalent) and three (7.0%) received immunosuppressive agents for the management of TRAEs. TRAEs resulted in treatment discontinuation in 13 patients (30.2%). There was no TRAEs-related death in the enrolled patients in this study during the follow-up period.

## 4. Discussion

Cancer-associated inflammation may contribute to the development of cancer and poor prognosis [17]. Tumor-associated neutrophils (TANs) are essential regulators of cancer-associated inflammation [30] and can influence genetic stability by releasing reactive oxygen species. They can induce tumor proliferation as well as immune escape by producing tumor necrosis factor, interleukin (IL)-1 and IL-6, and VEGF [30]. TANs are also involved in activating and forming mediators, such as neutrophil extracellular traps, essentially those involved in tumor progression [31]. Furthermore, TANs are largely responsible for the modulation of the tumor microenvironment, and an increased number of TANs may be primarily associated with treatment resistance [31]. Platelets protect tumor cells against immune elimination and accelerate trans-endothelial migration and metastasis, which plays a significant role in inducing rapid tumor progression [32]. Several platelet-secreted growth factors, such as vascular endothelial tumor growth, platelet-activating factor, and platelet-derived growth factor, influence tumor growth and metastasis [33]. Conversely, lymphocytes, particularly tumor-infiltrating lymphocytes (TILs), elicit an antitumor immune response, and elevated TIL counts in tumor tissues reportedly predict better oncological outcomes in patients with malignant neoplasms [33]. Therefore, NLR, PLR, and SII may be considered to be reflective of the balance between tumor and antitumor activities of the host immune response [17,20,21,22,23].

NLR is reportedly useful as an inflammatory biomarker for patients with mRCC [30,34,35]. Boissier et al. conducted a meta-analysis to assess the prognostic value of NLR in patients with RCC [34] and found that patients with a higher NLR had a poorer prognosis for metastasized or localized RCC and those with an NLR < 3 had a better OS and PFS [35]. According to Kobayashi et al., PFS was longer in patients with an NLR < 3.32 than in those with NLR ≥ 3.32 after first-line TKI therapy [30]. Furthermore, another study showed an NLR ≥ 4.0 to be an independent predictor for OS [35]. In our study, mRCC patients with an NLR < 2.8 had a significantly longer PFS compared with those with an NLR ≥ 2.8, suggesting a significant association between NLR and PFS. NLR may help predict outcomes of patients treated with NIVO + IPI for mRCC.

High PLRs and worse survival are correlated in patients with malignancies, such as colorectal, gastric, and non-small cell lung cancers [33]. Several recent meta-analyses have investigated whether PLR can predict outcomes of cancer patients treated with ICI therapy [36]. In 12 eligible studies (a total of 1340 patients with cancer), patients with a high PLR showed a significant association with a shorter OS and PFS (hazard ratio (HR) 2.02, *p* < 0.001 and HR 1.74, *p* < 0.001, respectively) compared with those with a low PLR [31]. Pooled analysis of 1528 patients with RCC collected from seven studies suggested that an elevation of PLR can effectively predict both OS (HR, 2.1; *p* = 0.001) and PFS (HR, 3.45; *p* = 0.001) [33]. Furthermore, a subgroup analysis of studies conducted in Asia found a high PLR to be significantly associated with worse OS and PFS in patients treated with targeted therapies for mRCC [33]. Herein, patients who had a high PLR before treatment of NIVO + IPI for mRCC were indicated as likely to have a poor PFS.

Patients with malignant tumors, including mRCC, reportedly exhibit a significant association between high SIIs and poor OS [19,23,37]. A systematic review and meta-analysis found a significant correlation between high SII and poor OS in patients with pancreatic carcinoma (HR: 1.43, 95% CI: 1.24–1.65, *p* < 0.001) [37]. Basal et al. have shown that survival in patients with mRCC differed significantly upon comparing those with low SII levels (<730) against those with high SII levels (≥730) (27.0 vs. 12.0 months, *p* < 0.001) [23]. In our study, patients showing high SIIs before treatment with NIVO + II for mRCC had poor PFS.

In our study, however, PFS was independent of the IMDC risk classification. For the IMDC system, risk classification was generated based on patients receiving first-line VEGF-targeted treatment for mRCC [24]. Although neutrophil and platelet counts were included in the IMDC risk classification, lymphocyte counts were not included [24]. Following the advent of ICIs, the treatment strategy has been markedly altered, and combination therapies involving ICIs and/or TKIs are the standard treatment for mRCC [2,9]. Therefore, new prognostic markers for mRCC are needed in this new era. NLR, PLR, and SII may potentially predict outcomes of patients treated with ICI therapy, particularly NIVO + IPI, for mRCC.

Based on the various types of cancer, several investigators have suggested a significant relationship between the development of TRAEs and the prognosis of patients treated with ICI therapy [38,39,40,41]. However, especially in mRCC, an extended analysis of the CheckMate 214 trial reported no significant difference in OS between patients with and without TRAEs following NIVO + IPI [42]. Conversely, Ikeda et al. reported that the development of TRAEs was an independent predictor of a longer PFS in patients with mRCC treated with NIVO + IPI (hazard ratio: 0.18, *p* = 0.0005) [43]. In our study, TRAEs was not significantly associated with PFS. Hence, future studies are needed to investigate the association between the development of TRAE and the oncological outcome of mRCC receiving ICI combination therapies.

When comparing NIVO + IPI with the ICI and TKI combination therapy, the PD rate is higher in NIVO + IPI [4,5,6]. However, in effective cases, it has been established that NIVO + IPI has a longer durable response and achieves therapeutic effects in a comparatively short period [11,42]. Furthermore, predictive factors, including NLR, PLR, and SII, before the treatment are relatively useful predictive factors of oncological outcomes. By using NLR, PLR, and SII in patients with a predicted poor prognosis, shortening the follow-up interval may aid in the early detection of disease progression and allow the patient to progress to the next treatment.

There are some limitations to this study. Being a multicenter retrospective study, a potential bias arising from diagnostic and therapeutic differences among participating institutions cannot be ruled out. Second, the relatively small sample size and short follow-up period might also influence the strength of our findings. Finally, this study did not address PD-L1 expression. Thus, to confirm our findings, prospective studies with large sample sizes and longer follow-up periods are required.

## 5. Conclusions

For patients treated with NIVO + IPI for mRCC, a significant association was noted between high NLR, PLR, and SII levels and poor PFS, indicating that NLR, PLR, and SII could be potentially used as biomarkers for predicting oncological outcomes, especially PFS, in patients with mRCC treated with NIVO + IPI. However, these findings should be validated further through prospective, large-scale, and long-term studies.

## Figures and Tables

**Figure 1 jcm-10-05325-f001:**
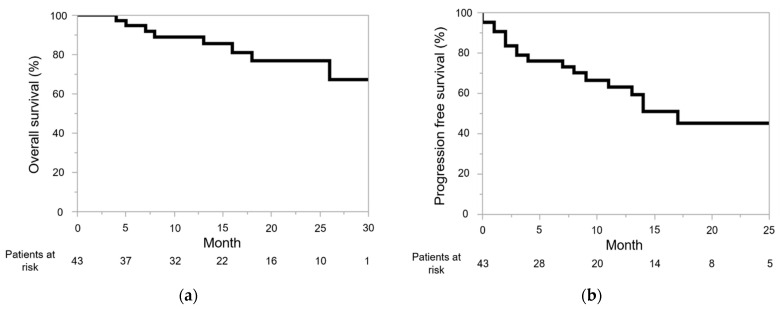
Kaplan-Meier analysis of overall survival (OS) (**a**) and progression-free survival (PFS) (**b**) in patients with metastatic renal cell carcinoma who received nivolumab and ipilimumab. OS at 6 and 12 months is 94.7% and 89.1%, respectively, while PFS is 76.1% and 63.1%, respectively.

**Figure 2 jcm-10-05325-f002:**
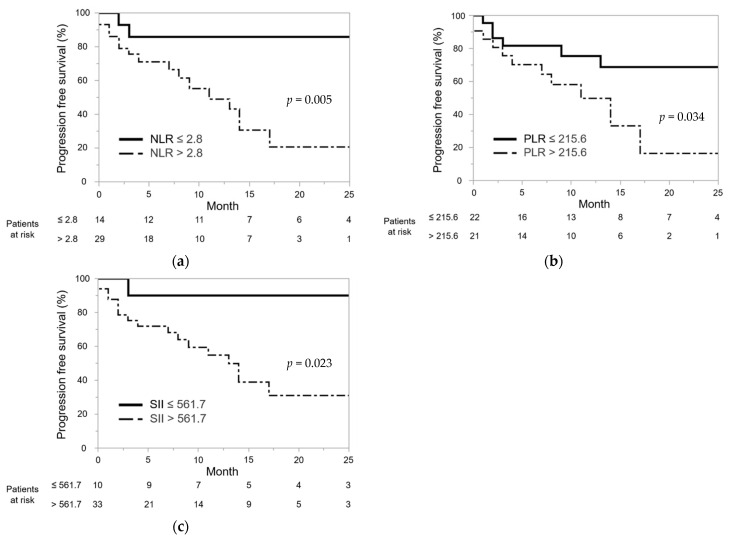
Kaplan-Meier curves illustrating progression-free survival (PFS): (**a**) per neutrophil-to-lymphocyte ratio (NLR) with a cutoff value of 2.8, (**b**) per platelet-to-lymphocyte ratio (PLR) with a cutoff value of 215.6, and (**c**) per systemic immune inflammation index (SII) with a cutoff value of 561.7. The 1-year PFS rates were 85.7% and 49.1% for NLR ≤ 2.8 and >2.8, respectively (*p* = 0.005; Figure 2a), and 75.5% and 49.7% for PLR ≤ 215.6 and >215.6, respectively (*p* = 0.034; Figure 2b). Regarding SII, the 1-year PFS rates were 90.0% and 54.8% when SII was ≤561.7 and >561.7, respectively (*p* = 0.023; Figure 2c).

**Table 1 jcm-10-05325-t001:** Demographic data for the enrolled patients.

Covariates	
Age (years, median, interquartile range)	69.0 (58.5–75.5)
Sex (number, %)	
Male	31 (72.1)
Female	12 (27.9)
Body mass index (kg/m^2^, median, interquartile range)	23.3 (20.6–25.5)
The Eastern Cooperative Oncology Group performance status (number, %)	
0	21 (48.8)
1	15 (34.9)
2	4 (9.3)
3	3 (7.0)
IMDC risk classification (number, %)	
Intermediate	25 (58.1)
Poor	18 (41.9)
Histology	
Clear cell renal cell carcinoma	27 (62.8)
Papillary renal cell carcinoma	1 (2.3)
Unknown	15 (34.9)
Neutrophil counts (*10^9^/L, median, interquartile range)	1.4 (1.1–1.8)
Lymphocyte counts (*10^9^/L, median, interquartile range)	4.5 (3.8–5.6)
Platelet counts (*10^9^/L, median, interquartile range)	265 (213–348)
Systemic immune inflammation index (median, interquartile range)	1045.2 (590.2–1862.2)
Neutrophil-to-lymphocyte ratio (median, interquartile range)	4.1 (2.7–5.9)
Platelet-to-lymphocyte ratio (median, interquartile range)	215.6 (144.7–316.3)
The patients who underwent surgery before administration of NIVO + IPI (number, %)	20 (46.5)
Number of metastatic sites	
0	4 (9.3)
1	12 (27.9)
2	14 (32.6)
≥3	13 (30.2)
Total number of metastatic sites (number, %)	
Lung	20 (46.5)
Lymph node	14 (32.6)
Bone	14 (32.6)
Liver	11 (25.6)
Brain	6 (14.0)
Adrenal gland	5 (11.6)
Pancreas	2 (4.7)
Local recurrence	2 (4.7)
Others	4 (9.3)

IMDC, International Metastatic Renal Cell Carcinoma Database Consortium; NIVO + IPI, combination nivolumab plus ipilimumab.

**Table 2 jcm-10-05325-t002:** Adverse events according to immunotherapy.

Event (number, %)	Any Grade	Grade 3/4
Treatment-related adverse events	31 (72.1)	19 (44.2)
Hypopituitarism	9 (20.9)	5 (11.6)
Maculopapular rash	7 (16.3)	0
Increased AST	5 (11.6)	5 (11.6)
Increased ALT	5 (11.6)	5 (11.6)
Colitis	4 (9.3)	3 (7.0)
Hypothyroidism	4 (9.3)	1 (2.3)
Pneumonitis	4 (9.3)	0
Pruritus	4 (9.3)	0
Arthritis	3 (7.0)	2 (3.5)
Myalgia	3 (7.0)	2 (3.5)
Weight loss	2 (3.5)	1 (2.3)
Hyperglycemia	1 (2.3)	1 (2.3)
Rheumatoid arthritis	1 (2.3)	1 (2.3)
Hyperthyroidism	1 (2.3)	0
Increased creatinine	1 (2.3)	0
Urticaria	1 (2.3)	0

ALT, alanine aminotransferase; AST, aspartate aminotransferase.

## Data Availability

Data and material are provided in this paper.

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
