# Peer review of "Utility of Neutrophil-to-Lymphocyte Ratio, Platelet-to-Lymphocyte Ratio, and Systemic Immune Inflammation Index as Prognostic, Predictive Biomarkers in Patients with Metastatic Renal Cell Carcinoma Treated with Nivolumab and Ipilimumab"

_jcm, 2021, doi:10.3390/jcm10225325_

Round 1

Reviewer 1 Report

In the present manuscript, the authors investigate the role of circulating inflammatory parameters for prediction of outcome in mRCC treated with immune-checkpoint inhibitors. The argument is up-to-date, although the correlation performed in the article is not completely new. Yet, the implication for combination immunotherapy (NIVO + IPI) is of great interest.

Minor revision is required (see comment below).

TRAEs are typically correlated with outcome in immunotherapeutic regimens. Almost half of the patient cohort presented TRAEs in the present study. Therefore, it would be of interest to include the results of the statistical association of TRAEs and PFS in mRCC treated with NIVO+IPI. Please change accordingly the methods and provide results. Discussion should be adjusted as a consequence.

Author Response

Responses to the reviewer's comments

We would like to thank the Reviewers for taking the time and effort necessary to review the manuscript. We sincerely appreciate all the valuable comments and suggestions, which helped us to improve the quality of the manuscript.

Response to Reviewer 1

The authors appreciate the reviewer’s comments. The authors’ point-by-point responses to the comments are given below.

  1. TRAEs are typically correlated with outcome in immunotherapeutic regimens. Almost half of the patient cohort presented TRAEs in the present study. Therefore, it would be of interest to include the results of the statistical association of TRAEs and PFS in mRCC treated with NIVO+IPI. Please change accordingly the methods and provide results. Discussion should be adjusted as a consequence.

Response:

We have added the results of statistical association of TRAEs and PFS on line 160.  

Additionally, TRAEs were not significantly associated with PFS (p = 0.085).

We also adjusted the methods on line 125, and discussion on line 241.

In addition, the patients were divided into two groups with TRAEs ≥ Grade 3 and ≤ Grade 2.

Based on the various types of cancers, several investigators have suggested a significant relationship between the development of TRAEs and the prognosis of patients treated with ICI therapy [38-41]. However, especially in mRCC, an extended analysis of the CheckMate 214 trial reported no significant difference in OS between patients with and without TRAEs following NIVO+IPI [42]. Conversely, Ikeda et al. reported that the development of TRAEs was an independent predictor of a longer PFS in patients with mRCC treated with NIVO+IPI (hazard ratio: 0.18, p = 0.0005) [43]. In our study, TRAEs was not significantly associated with PFS. Hence, future studies are needed to investigate the association between the development of TRAE and the oncological outcome of mRCC receiving ICI combination therapies.

Reviewer 2 Report

This study addresses a current topic.

The manuscript is quite well written and organized. English could be improved.

Figures and tables are comprehensive and clear.

The introduction explains in a clear and coherent manner the background of this study.

We suggest the following modifications:

  • Introduction section: although the authors correctly included important papers in this setting, we believe some studies should be cited within the introduction (PMID: 32799582; PMID: 33246931; PMID: 34265504;  ), only for a matter of consistency. We think it might be useful to introduce the topic of this interesting study
  • Methods and Statistical Analysis: nothing to add.
  • Discussion section: Very interesting and timely discussion. Of note, the authors should expand the Discussion section, including a more personal perspective to reflect on. For example, they could answer the following questions – in order to facilitate the understanding of this complex topic to readers: what potential does this study hold? What are the knowledge gaps and how do researchers tackle them? How do you see this area unfolding in the next 5 years? We think it would be extremely interesting for the readers.

However, we think the authors should be acknowledged for their work. In fact, they correctly addressed an important topic in renal cell carcinoma, the methods sound good and their discussion is well balanced.

One additional little flaw: the authors could better explain the limitations of their work, in the last part of the Discussion.

We believe this article is suitable for publication in the journal although major revisions are needed. The main strengths of this paper are that it addresses an interesting and very timely question and provides a clear answer, with some limitations.

We suggest a linguistic revision and the addition of some references for a matter of consistency. Moreover, the authors should better clarify some points.

Author Response

Responses to the reviewer's comments

We would like to thank the Reviewers for taking the time and effort necessary to review the manuscript. We sincerely appreciate all the valuable comments and suggestions, which helped us to improve the quality of the manuscript.

Response to Reviewer 2

The authors appreciate the reviewer’s comments. The authors’ point-by-point responses to the comments are given below.

  1. Introduction section: although the authors correctly included important papers in this setting, we believe some studies should be cited within the introduction (PMID: 32799582; PMID: 33246931; PMID: 34265504), only for a matter of consistency. We think it might be useful to introduce the topic of this interesting study

Response:

We have cited some studies in introduction and added following sentences on line 53.

In the final oncological outcomes of Checkmate 214 after the long-term follow-up period, NIVO+IPI continued to demonstrate durable efficacy benefits compared with sunitinib [11]. Massari et al. reported that the clinical benefit obtained by patients with mRCC who received immunotherapy combinations with a complete response (CR) rate was more than in those who received the TKI and ICI combination [12].

We have added the following references:

  1. Massari F, Mollica V, Rizzo A, Cosmai L, Rizzo M, Porta C. Safety evaluation of immune-based combinations in patients with advanced renal cell carcinoma: a systematic review and meta-analysis. Expert Opin Drug Saf. 2020, 19(10), 1329-1338.
  2. Albiges L, Tannir NM, Burotto M, McDermott D, Pilmack ER, Barthélémy P, et al. Nivolumab plus ipilimumab versus sunitinib for first-line treatment of advanced renal cell carcinoma: extended 4-year follow-up of the phase III CheckMate 214 trial. ESMO Open. 2020, 5(6), e001079.
  3. Massari F, Rizzo A, Mollica V, Rosellini M, Marchetti A, Ardizzoni A, et al. Immune-based combinations for the treatment of metastatic renal cell carcinoma: a meta-analysis of randomised clinical trials. Eur J Cancer. 2021, 154, 120-127.
  1. Discussion section: Very interesting and timely discussion. Of note, the authors should expand the Discussion section, including a more personal perspective to reflect on. For example, they could answer the following questions – in order to facilitate the understanding of this complex topic to readers: what potential does this study hold? What are the knowledge gaps and how do researchers tackle them? How do you see this area unfolding in the next 5 years? We think it would be extremely interesting for the readers.

However, we think the authors should be acknowledged for their work. In fact, they correctly addressed an important topic in renal cell carcinoma, the methods sound good and their discussion is well balanced.

Response:

We agreed about reviewer comments. We have added following sentences in discussion on line 241.

Based on the various types of cancers, several investigators have suggested a significant relationship between the development of TRAEs and the prognosis of patients treated with ICI therapy [38-41]. However, especially in mRCC, an extended analysis of the CheckMate 214 trial reported no significant difference in OS between patients with and without TRAEs following NIVO+IPI [42]. Conversely, Ikeda et al. reported that the development of TRAEs was an independent predictor of a longer PFS in patients with mRCC treated with NIVO+IPI (hazard ratio: 0.18, p = 0.0005) [43]. In our study, TRAEs was not significantly associated with PFS. Hence, future studies are needed to investigate the association between the development of TRAE and the oncological outcome of mRCC receiving ICI combination therapies.

When comparing NIVO+IPI with the ICI and TKI combination therapy, the PD rate is higher in NIVO+IPI [4-6]. However, in effective cases, it has been established that NIVO+IPI has a longer durable response and achieves therapeutic effects in a comparatively short period [11,42]. Furthermore, predictive factors, including NLR, PLR, and SII, before the treatment are relatively useful predictive factors of oncological outcomes. By using NLR, PLR, and SII, in patients with a predicted poor prognosis, shortening the follow-up interval may aid in the early detection of disease progression and allow the patient to progress to the next treatment.

We have added the following references:

  1. Fiala O, Sorejs O, Sustr J, Kucera R, Topolcan O, Finek J. Immune-related Adverse Effects and Outcome of Patients With Cancer Treated With Immune Checkpoint Inhibitors. Anticancer Res. 2020, 40(3), 1219-1227.
  2. Weber JS, Hodi FS, Wolchok JD, Topalian SL, Schadendorf D, Larkin J, et al. Safety Profile of Nivolumab Monotherapy: A Pooled Analysis of Patients With Advanced Melanoma. J Cin Oncol. 2017, 35(7), 785-792.
  3. Cortellini A, Chiari R, Ricciuti B, Metro G, Perrone F, Tiseo M, et al. Correlations Between the Immune-related Adverse Events Spectrum and Efficacy of Anti-PD1 Immunotherapy in NSCLC Patients. Clin Lung Cancer. 2019, 20(4), 237-247.
  4. Sanz-Segura P, García-Cámara P, Fernández-Bonilla E, Arbonés-Mainar JM, Bernal Monterde V. Gastrointestinal and liver immune-related adverse effects induced by immune checkpoint inhibitors: A descriptive observational study. Gastroenterol Hepatol. 2021, 44(4), 261-268.
  5. Motzer RJ, Escudier B, McDermott DF, Arén Frontera O, Melichar B, Powles T, et al. Survival outcomes and independent response assessment with nivolumab plus ipilimumab versus sunitinib in patients with advanced renal cell carcinoma: 42-month follow-up of a randomized phase 3 clinical trial. J Immunother Cancer. 2020, 8(2), e000891.
  6. Ikeda T, Ishihara H, Nemoto Y, Tachibana H, Fukuda H, Yoshida K, et al. Prognostic impact of immune-related adverse events in metastatic renal cell carcinoma treated with nivolumab plus ipilimumab. Urol Oncol. 2021, 39(10), 735.e9-735.e16.
  1. One additional little flaw: the authors could better explain the limitations of their work, in the last part of the Discussion.

Response:

We have added following sentences on line 263.

Thus, to confirm our findings, prospective studies with large sample sizes and longer follow-up periods are required.

Round 2

Reviewer 2 Report

The authors modified the manuscript according to our suggestions.

We recommend Acceptance.